

# North Atlantic subpolar gyre along predetermined ship tracks since 1993: a monthly dataset of surface temperature, salinity, and density.

**Gilles Reverdin[1], Hedinn Valdimarsson[2], Gael Alory[3], Denis Diverres[4] , Francis Bringas[5], Gustavo Goni[5], Lars Heilmann[6], Leon Chafik[7], Tanguy Szekely[8], Andrew R. Friedman[9]**

[1] LOCEAN, Sorbonne-Université, CNRS, IRD, MNHN, Paris, France

[2] Marine and Freshwater Research Institute, Reykjavik, Iceland

[3] LEGOS, Université de Toulouse, UPS, CNRS, IRD, CNES, Toulouse, France

[4] US IMAGO, IRD, Brest, France

[5] National Oceanic and Atmospheric Administration/Atlantic Oceanographic and Meteorological Laboratory, Miami, FL, USA

[6] Greenland Institute of natural resources, Nuuk, Greenland, DK

[7] Geophysical Institute, University of Bergen and Bjerknes Centre for Climate Research, Bergen, Norway

[8] IUEM, UMS3113, UBO-CNRS-RD, Plouzané, France

[9] School of Geosciences, University of Edinburgh, Edinburgh, Scotland, UK.



**Abstract**
We present a binned product of sea surface temperature, sea surface salinity and sea
surface density data in the North Atlantic subpolar gyre for the 1993-2017 that resolves
seasonal variability along specific ship routes (
https://dx.doi.org/10.6096/SSS-BIN-NASG). The characteristics of this product are
described and validated through comparisons to other monthly products. Data presented
in this work was collected in regions crossed by two predetermined ship transects,
between Denmark and western Greenland (AX01) and between Iceland, Newfoundland,
and the northeastern USA (AX02).  The analysis and the strong correlation between
successive seasons indicate that in large parts of the subpolar gyre, the binning approach
is robust and resolves the seasonal time scales, in particular after 1997 and in regions
away from the continental shelf. Prior to 2002, there was no winter sampling over the
west Greenland shelf. Variability in sea surface salinity increases towards Newfoundland
south of 54°N, as well as in the western Iceland Basin along 59°N. Variability in sea
surface temperature presents less spatial structure with an increase westward and towards
Newfoundland. The contribution of temperature variability to density dominates in the
eastern part of the gyre, whereas the contribution of salinity variability dominates in the
southwestern part along AX02.
**Copyright statement**
The author's copyright for partner 5 of this publication is transferred to the National
Oceanic and Atmospheric Administration (NOAA) (for FB and GG).
**Data availability**
The gridded data set is freely available and accessible at
https://dx.doi.org/10.6096/SSS-BIN-NASG
The XBT data collected along AX01 and AX02 is available at
http://www.aoml.noaa.gov/phod/hdenxbt




## 1. Introduction

The North Atlantic Subpolar Gyre (NASG) has been extensively studied and observed
during the last 25-years. This period presents the succession of a cold period in the early
1990s associated with strong North Atlantic Oscillation (NAO) forcing, a warmer period
in 2000-2009, followed by a cooling (Robson et al., 2016), and strong NAO forcing in
2014 and 2015 (Josey et al., 2017). These conditions were associated with strong
variability in intermediate water formed in the Labrador Sea, south-western Irminger Sea
or South of Greenland, with strong formation years following strong atmospheric and
NAO forcing years (Yashayaev and Loder, 2016; Fröb et al., 2015; de Jong et al., 2016,
Piron et al., 2017). There has also been extensive variability in mode waters and their
thickness in the northern or northeastern subpolar gyre, such as the Reykjanes mode
water (Thierry et al 2008) or the Rockall Trough mode water (Holliday et al., 2015). The
changes in these subsurface water properties and distributions drive ocean circulation and
in particular of the Atlantic Meridional Overturning Oscillation (AMOC) variability
(Robson et al., 2016; Rahmstorf et al., 2015). The surface layer provides the link between
the ocean interior and the atmosphere.

Surface variability in oceanic properties responds to atmospheric forcing and ocean
circulation changes. In particular, NAO is known to strongly influence heat and
freshwater fluxes in this region (Cayan, 1992; Hurrell et al., 2013; Bojariu and Reverdin,
2002) and thus sea surface temperature (SST) and sea surface salinity (SSS) (Josey and
Marsh, 2005). Changes in freshwater fluxes from continental run-off and ice melt are also
expected to change surface properties in the NASG (Böning et al., 2016). Net run-off
from Greenland has considerably changed during the last decades (van der Broeke et al.,
2016).The role of changes in ocean circulation have also been identified. For instance, the
proportion of inflowing subtropical water was found to have increased in the 1995-2005
period compared to the previous two decades (Häkkinen et al., 2011; Häkkinen, 2013),
followed by a net reduction of this input (Robson et al., 2016), which could have
contributed to the more recent decadal cooling/freshening (see also Piecuch et al., 2017).
A strong cold blob and anomalous cooling (and freshening) area has appeared in the



center of the NASG since the late-2000, and has also been linked indirectly to changes in
the AMOC (Rahmstorf et al., 2015; Josey et al., 2017)

It has been speculated that the changes in atmospheric conditions, and of the resulting
central gyre temperature and density associated with the strength of the gyre circulation
are associated with zonal displacements of the subpolar front (Hatun et al., 2005;
Sarafonov, 2009). This has been disputed (Foukal and Lozier, 2017), and has not been
clearly identified in subsets of in situ current measurements along 59°N in two multi-year
periods (Rossby et al., 2017), although recent analysis of altimetric sea level data also
support an eastward displacement of the subpolar front during the recent period of strong
atmospheric forcing (Zunino et al., 2017; their Fig. 8). The strong changes in thermocline
and water masses associated with the fronts have been used by Stendardo et al (2017) to
reconstruct surface temperature and salinity based on satellite altimetry data, which also
suggests displacements of the subpolar front as a result of NAO forcing. However, this
method does not work in the interior of the NASG.

Here, we present an effort to construct monthly time series of temperature (T), salinity
(S) and density along tracks in the interior of the gyre. The data and methods used are
first described in section 2, then the time series are presented in section 3. Basic
characteristics are provided: interannual standard deviations and an EOF analysis of
interannual variability. The characteristics of the data validation are presented in the
appendices.

**2. Data and Methods**
**2.1 Data**
A large part of the data presented here are from SBE21 and SBE45 thermosalinographs
(TSG) installed on ships running along the AX01 transect between Denmark and western
Greenland and along the AX02 transect between Iceland, Newfoundland and the north-
eastern USA (Fig. 1). Along AX01, TSG data were collected on M/V Nuka Arctica
between July 1997-2017 (with intake temperature in 2005-2017). Along AX02, TSG data



are available between April 1994 and December 2007 and between March 2011 and
March 2016 (with intake temperature during April 1994-1996).

The first installation on Nuka Arctica was done on a pumped water circuit in the bow
thruster room of the ship, with little warming, but frequent interruptions during and after
bad weather. In 2006, it was moved to the engine room at approximately mid-ship,
roughly 5-m below the water line. During some winters (January-March 1997-2002)
there were no cruises on this ship. The most common route crosses the North Atlantic
subpolar gyre along 59°-59.5°N (B-AX01), but the ship has often taken a different route,
in particular further north (N-AX01) (see e.g. Chafik et al., 2014, their Fig. 1). Along
west Greenland, at least north of 62°N, the route is fairly repeated between transects and
often runs in mid-shelf between ports-of-call, very often up to southern Disko Bay (G-
AX01), with a few crossings in summer further north to Thule in northwest Greenland.

Along AX02, a succession of ships has been used, with different installations usually in
the engine room at mid-ship, between 4 and 7 m below the water line. The route taken by
these vessels is often roughly straight between southeastern Newfoundland and the
western tip of the Reykjanes peninsula (Fig. 1, what we will refer as the standard route B-
AX02), but with some deviations depending on sea ice or weather conditions. Due to
seasonal sea ice, in particular, there were no standard TSG data on the route north-east of
Newfoundland on shelf and slope in February-April 1994-1995 and 2014-2016.

The validation and correction of the TSG salinity data is mostly based on comparison
with water samples collected from a water intake at the TSG (AX01 and AX02) and
using nearby upper level of Argo float data (primarily for AX02) (Alory et al., 2015). On
AX02, adjusting T from the TSG to near-surface ocean temperature was done when no
intake measurements were available, largely based on comparison with T from
Expandable bathythermograph (XBT) observations at 5-7 m. XBT observations along
these two transects were started in 2000 (AX01) and 1993 (AX02) and have produced
approximately 4,000 temperature profiles available for these comparisons. In addition T
from the TSG on Nuka Arctica (AX01) were also used to adjust T along AX02 where



124 there were crossovers of the AX01 and AX02 ships route. Validation of T and S data

125 from the thermosalinographs is discussed further in App. A. For AX02, additional T and

126 S data originate from seasonal surface sampling, in particular in July1993, January and

127 April 1994, in 1995, and in 2007-2017. TSG data from several research cruises were also

128 included. Upper-level data (near 5-7 m) of profiles from Argo, earlier PALACE floats

129 (since 1996) and from CTD casts were also considered, as well as data from drifters

130 equipped to measure precise temperature and salinity.

131

132 **2.2 Methods**

133 We construct monthly binned T, S, and density time series starting in mid-1993 along

134 two standard sections intersecting near 59.5°N/32°W: B-AX01 between the North Sea

135 and South Greenland and B-AX02 between Iceland and southern Newfoundland (Fig. 1).

136 The B-AX01 section extends from the south-east of Cape Farewell (excluding the shelf

137 or its vicinity) to the northwestern North Sea north-east of Scotland over the shelf. A

138 separate binning G-AX01 is done on the Greenland shelf between southern Disko Bay

139 (near 68.2°N/54°W) and northwest of Cape Farewell, but only since July 1996 (with no

140 data north of 64.5°N in 1996-1997). We also binned data on an alternate route often used

141 by Nuka Arctica across the Irminger Sea and Iceland Basin (N-AX01), to the north of B-

142 AX01. For B-AX02, we include two bins over the Newfoundland shelf and two bins

143 further to the north-east over the continental slope, followed by more regular bins along

144 the standard section.

145

146 First, a gridded seasonal cycle is subtracted from the data to create anomalies that are

147 then grouped in the bins on a monthly time scale. The average seasonal cycle is based on

148 120-year of data in the NASG (Friedman et al., 2017), and is on a 0.5° x 1° latitude x

149 longitude grid. After creating the time series, the average seasonal cycle is modified and

150 adjusted over the time series length to bring it back to no average anomalies. The actual

151 average salinity is also provided, by adding this average seasonal cycle. Time series

152 along B-AX01 contain some long-lasting data gaps until late 1997 that were filled with

153 data along 58°N or 60°N, therefore larger errors attributed. Along, G-AX01, there are no

154 winter data (January- March) in 1997-2002. Time series along B-AX02 start in July 1993

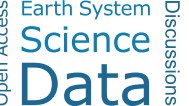

with few gaps longer than three months, the longest being associated to winters with ice
presence over the Newfoundland shelf or slope. The time series are then smoothed by a
1-2-1 running-mean over successive months. Before performing an empirical orthogonal
function analysis (EOF), gaps in the time series are filled by first linearly interpolating
from neighboring spatial bins, and then in time from neighboring time steps. They are
then normalized to unit variance. Comparison of this gridded product against other
gridded products is provided in App. B.

**3. Results**
**3.1 variability along AX01 and AX02**
The Hovmøller diagram of seasonal salinity anomalies are presented on Fig. 2. A rather
similar variability is portrayed where the two sections B-AX01 and B-AX02 intersect,
although clearly B-AX01 indicates a strong longitude dependence of the signals
portrayed just to the east of the intersection of B-AX02.

The B-AX02 salinity plot (Fig. 2, top right) suggests large spatial variations characterized
by interannual to decadal variability. On the shelf and slope regions, in particular near
Newfoundland, there seems to be more short-term variability. However, in these regions,
error estimates are also larger, to some extent as a result of insufficient sampling, as well
as due to unresolved high-frequency variability. This results in weak correlation of S
anomalies between successive seasons (three months apart), although there is a tendency
for negative low frequency anomalies until 2000 and since 2010. Correlation in other
regions of AX02 between successive seasons is larger (correlation coefficient at least
0.6), indicating a dominance of interannual and lower frequency variability over intra-
annual variability. There is less spatial variability along B-AX01 (Fig. 2, top left). There
is a tendency for differences across the Reykjanes ridge, such as in 1994 or 2015-2017,
with large negative anomalies in the western Iceland Basin. Variability on European
shelves tend also to be different. On B-AX01, correlation is also large between
successive seasons, with an exception in the last 200 km from the shelf break off southern
Greenland. There, however, the TSG transects do not resolve well enough the spatial and
temporal variability.




Temperature anomalies (Fig. 2, middle panel) tend not to be correlated with the salinity
ones, although there is some suggestion that the decadal variability is correlated (except
on the shelves). This is seen here as the negative SST anomalies near the beginning and
end of the time series with warmer temperatures in the 2000-2009 period, roughly
corresponding to SSS variability of the same sign. Variability is slightly larger along B-
AX02, as expected from the known westward increase in SST variability portrayed for
example in the Hadley Centre SST data set (HADSST3). Altogether there is not a large
spatial variability in the temperature signals along these transects, at least on seasonal or
longer time scales, except for some differences on the southern part of B-AX02 compared
to other regions.

Density anomalies (Fig.2, lower panel) are a result of both temperature and salinity
anomalies. Except in the southern part of B-AX02 (south of 54°N), temperature
variability tends to have a larger contribution than salinity variability to density (in
particular east of the Reykjanes Ride or north of 60°N). Thus, as for T, density anomalies
along B-AX01 tend to present small longitudinal variations, with in particular highest
positive density anomalies in the first few years and in mid-2014 to early-2016. Since
early 2016, negative density anomalies are confined east of the Reykjanes Ridge. Along
B-AX02, there is a larger contrast, with a transition near 52-54°N, with the density
anomalies looking more like S south of it and more like T north of it. The correlation
between density anomalies in successive seasons is also smaller for surface density than
for T and S.

To a large extent, section N-AX01 (Fig. 3) presents variability that is coherent with what
is seen on B-AX01 along 59°N (Fig. 2). However, whereas in the Iceland Basin near 10-
18°W along N-AX01, one also finds the freshening happening by mid-2015, further west
(and closer to Iceland) as well as in the northeastern Irminger Sea east of 35°W, the
freshening happens later in 2016 and 2017 (with some suggestion of a weaker winter
signal). This did not show up further south along 59°N in the eastern Irminger Sea away
from the Reykjanes Ridge until late 2017 (Fig. 2). Along N-AX01, at 20°W, very close to



southern Iceland, there are also isolated patches of larger anomalies, possibly related to
local freshwater inputs from Iceland. To the east of the section, the last bin near the
Shetlands Islands portrays a variability often very close to what is found further west in
the deeper Ocean, whereas the two easternmost bins of the section along 59°N on the
shelf (northwest and northeast of Scotland) seem to present a different variability.

Finally, variability on the west Greenland southwestern shelf (Fig. 3) is rather different
for S than to the east in the Irminger Sea along B-AX01 or N-AX01. Except for the most
southern box to the west of Cape Farewell, variability in S is rather coherent
meridionally. For example, negative anomalies are observed in 2000, from mid-2006 to
early 2009, and even more in 2010-2013 with a peak in the second half of 2012, and
positive anomalies in 2015-2017. The extreme negative S values in late 2012 are
consistent with the outstanding Greenland sheet melt that occurred that year (van der
Broeke et al., 2016; Fettweis et al., 2017). On the other hand, other years with very large
southern Greenland ice sheet melt (1995, 2002, 2005-2007, 2010, 2011) do not show as
well in surface salinity. Temperature variability tends to be also of the same sign along
the section, but with some notable exceptions. For instance, negative T anomalies are
found in 2015-2017 north of 65°N, and not further south.

**3.2 Interannual RMS variability**
For each month of the calendar year, we evaluate the interannual RMS variability for
each spatial bin. This gives us an estimate of the seasonal cycle of the interannual RMS
variability (Fig. 4). For S (Fig. 4, top panels), large RMS values are found on the
southern part of B-AX02 with a large decrease between 52°N and 54°N (52 and 53°N in
winter). Near 55°N, there is minimum variability during winter-spring, then increases
again near 57-59°N, followed by a strong decrease towards Iceland. RMS variability in S
presents a seasonal cycle with a spring minimum over the Newfoundland shelf, which is
less noticeable along the continental slope. Further offshore and until 54°N, there is a
minimum variability in spring (and maximum during late summer/autumn). North of
54°N, there is a winter to late winter minimum although very weak near 56-59°N. This
winter to early spring minimum is also very prominent along N-AX01, except in western



Irminger Sea, close to the Greenland shelves (not shown). Along B-AX01 at 59°N for S,
the maximum RMS variability is found in the western Iceland Basin (20-30°W), then less
further east (as well as in the Irminger Sea). There are also larger RMS east of 10°W
along shelves/north-western North Sea (and last eastern box of N-AX01 near the
Shetlands). There is not much seasonal variability in RMS along 59°N, although with
weaker RMS in winter-early spring in the Irminger Sea.

For T (Fig. 4, middle panels), larger variability is found south of 54°N towards
Newfoundland (except on the shelf, where winter SST variability is lower). Along 59°N,
larger variability is found in the western Irminger Sea and eastern Iceland Basin. There is
a seasonal modulation of RMS values with larger values in June-July north of 54°N along
B-AX02 and along 59°N. In the western Irminger Sea or south of 54°N closer to
Newfoundland, maximum RMS is shifted later in July to early autumn.

The surface density RMS seasonal cycle (Fig. 4, lower panels) is a mix of what is seen on
temperature and salinity.  Along B-AX01 and N-AX01, density variations are dominated
by temperature variations, except west of 40°W along B-AX01 and close to Iceland,
where S and T have comparable contributions. Along B-AX02, south of 54°N, salinity
contributes more to density variability than temperature, whereas further north, the two
contributions are of a similar magnitude.

**3.2 EOF analysis**
When performing an EOF analysis on S, on B-AX01 and B-AX02 together, little
seasonal dependence is observed in the first two components**:** similar time series are
almost found when performing the EOF analysis on the whole time series or on low
passed filtered time series for different seasons (not shown). The associated principal
components are very similar for the two tracks, thus we jointly analyzed the two gridded
data sets after low-pass filtering by the 15-month running mean filter (Fig. 4). The
principal components associated with EOF1 and EOF2 both present a large variability at
periods of 5 years or more. PC1 largest negative anomalies are in 1994-95 and in 2016-
2017, and largest positive values in 2004 and 2009, whereas PC2 largest negative values



are in 2015, but with an apparent trend superimposed. PC1 resembles the sea surface
height variability in the northern North Atlantic and hence gyre variability (Chafik et al.,

281 2018).


EOF1 has large positive values across the two sections, except in the far west of AX01
(close to the east Greenland Current), and on the Labrador shelf. EOF2, which overall
explains only 14% of the variance, has positive values both in the Labrador Sea (B-AX02
south of 53°N) and in the western Iceland Basin (27-17°W) along B-AX01 (to a smaller
scale), with negative values in the Irminger Sea along AX01 peaking near 40°W.
Maximum values (where it is positive) never explain more than 50% of the local
variance. EOF3 (10% of the variance) has large values only over the Labrador shelf and
slightly north of it until 55°N along B-AX02, and seems to correspond to higher
frequency variability.

**Data availability**
The gridded data set is freely available and accessible at
https://dx.doi.org/10.6096/SSS-BIN-NASG
The XBT data collected along AX01 and AX02 is available at
http://www.aoml.noaa.gov/phod/hdenxbt
**4. Conclusion**
The validated data presented here are able to characterize the seasonal variability of
surface temperature and salinity along two transects crossing the North Atlantic subpolar
gyre (along 59°N and from south-west Iceland to south-east Newfoundland) from July
1993 to December 2017. The time series presented here describe the interannual
variability at seasonal resolution over this 21 to 25-year period except for some winter
gaps over the Newfoundland shelf and along west Greenland, as well as until 1996 in the
Iceland Basin along 59°N, and until mid-1997 along parts of west Greenland. To describe
this variability these time series are better than current SST or SSS ocean data gridded
analyses such as those provided in EN4 or CORA, in particular before the Argo period.
These time series provide added information, in particular on the shelves and continental
slope regions that is not available from Argo float data, despite Argo reaching nominal

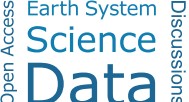

density since the early 2000s. Also, they are complementary to indirect analyses of the
variability based largely on satellite altimetry ( Stendardo et al., 2016), which only work
as long as a strong relationship between dynamic height, sea level and surface T and S
exist, such as near fronts in the open ocean (Dong et al., 2015). This excludes most of the
area investigated here.

In the interior of the subpolar gyre, the time series can be used to precisely monitor the
arrival of very large freshwater salinity anomalies in recent years, and to characterize
how they relate or not with temperature anomalies. They also suggest similarities with an
earlier event in 1994-1996, which is unfortunately not as well sampled overall (Reverdin
et al., 2002). The salinity time series are rather different on the shelves sampled here, in
particular west of Greenland and near Newfoundland. This is expected, because of the
different water masses with a large proportion of water advected from the Arctic or
influenced by continental inputs. Sampling with the ships of opportunity is not always
sufficient in these areas, due to the presence of seasonal sea ice, and would need to be
complemented by other observational platforms.

In some areas, such as on the shelves or south of 54°N along B-AX02, there is a seasonal
modulation of surface salinity variability. In most areas, salinity variability tend to be
largest in summer or early autumn, although there are areas, such as along 59°N, with a
weak seasonal cycle of this variability. Further interpretation of these data would require
at least contemporary information on air-sea fluxes (heat, fresh water), mixed layer depth,
and ocean circulation.

Results and data presented here highlight the importance of repeated ocean observations
from volunteer ships, and the value of complementary data to better assess and monitor
the state of the ocean and its variability from seasonal to interannual time scales.

**Appendix A: Validation of TSG data**
TSG observations from M/V Nuka Arctica form the core of the B-AX01, N-AX01 and
G-AX01 are available since 1997. The salinity values were validated and adjusted using



mostly surface water samples following Alory et al. (2015). An intake temperature
measurement was used since late 2004 to adjust the temperature measurements reported
by the TSG. Before that, ad hoc adjustment was made on Nuka Arctica TSG temperature
based on comparison with nearby data, but showing often very small differences, of less
than 0.1°C.

We checked the consistency of these T-S data of Nuka Arctica with other upper ocean
data. The TSG temperature and salinity data do not present significant biases with the
upper level of Argo profiles, close to 5-8 m depth. The average differences (TSG-Argo)
in T of 0.03°C and in S of 0.01 psu are compatible with 0 at the 95% level (based on 226
profiles within 50 km and 5 days of ship's track, accepting differences of 1°C and of 0.2
psu, which removes 11% of outliers).

The 'adjusted' temperature reported by the TSG was also compared with the temperature
of the XBTs launched usually every three months from the *Nuka Arctica* since 2001
(Rossby et al., 2017). The comparison was done with XBT temperature at 7-m depth. We
first average the comparisons over individual transects and estimate a mean and RMS
difference. Then we average these transect summaries. When removing 5 transects for
which there is too large a scatter in the individual matches (RMS difference larger than
0.2°C), the average temperature difference for 40 transects is -0.056°C with an RMS
difference between individual transect summaries of 0.075°C (if individual transects were
independent and in a Gaussian distribution, this would result in a 95% percentile range
between -0.032 and -0.080°C). This average difference fits with the expected near surface
temperature warm bias of XBTs for those years (Reverdin et al., 2009). The five
occurrences with larger scatter fall in two categories: two in early June in the eastern part
of the section with weak wind and a very likely stratification near the surface, resulting in
T from the TSG higher than T from the XBT profiles at 7m, and three where the flow rate
was very weak (in 2001-2003). With the TSG placed in the bowhead of the ship until
2005, it is unlikely that T measured during those transects would present large biases
with respect to outside SST, although clearly there is a time lag and time integration of
the ocean temperature in those records. Because data at large spatial scales seemed



reasonable during these weak-flow instances, including the 5 events does not change
significantly the average bias. Thus, we retained these data in the data set, despite the
likely time delay. In summary, although there can be errors on individual transects, the
comparisons suggest high consistency between TSG data and other validated data a few
meters below the surface.

We carried a similar comparison for TSG data AX02 data since 1994, but although
average results are similar, scatter is larger. The comparisons are also more difficult to
interpret, because of many changes in how and where the TSGs were installed on
different ships during the 1994-2016 period, frequent insufficient flow through the
instrument, and also because XBT and Argo data were used to adjust the TSG
temperatures when there was no intake temperature measurements. Notice also that for 6
crossings (in July 1993, January and April 1994, as well as in 2016-2017), temperature
was measured by the bucket method, taking care of leaving the bucket long enough in the
sea and measuring T quickly (within 30 seconds) after retrieving the bucket. The data
were compared for two crossings with intake temperature measurements, suggesting
small negative biases (at most -0.1°C), except during high wind conditions, which were
not frequent.

**Appendix B: comparison with ENACT, CORA and Armor3D gridded products**
Mapped analysis products of the hydrographic data sets EN4 and CORA6.1 are based on
objective mapping (Good et al., 2013 for EN4 and Cabanes et al., 2013, Gaillard et al.,
2013 for CORA/ISAS), and contain a level near the surface which is used here. Mapped
products from Armor3D are largely based on altimetric sea level data with T and S
adjusted to in situ T and S profiles (Guinehut et al., 2012).

We compare the binned (B-AX01) monthly time series (59-60°N) (left panels of fig. 4) to
interpolated EN4, CORA6.1 and Armor3D products at the same sites and with additional
1-2-1 smoothing applied over successive months (EN4-AX01, CORA-AX01, Armor3D-
AX01) (June 1993 to December 2015).  The results are summarized by presenting
longitude sections of correlation and RMS variability (Fig. B1). For S, there is little



correlation in SSS with EN4, except in the western Iceland Basin, and RMS variability is
much higher in EN4 surface fields (often by a factor of 2). Amplitudes are closer in
CORA-AX01 and Armor3D-AX01, although there are smaller than those observed in the
eastern Irminger Sea, and correlation is high except near the slopes. Interestingly, when
averaging vertically EN4 salinity over the 0-500m layer, correlation with B-AX01
strongly increases everywhere (with coefficients often larger than 0.6), and becomes
significant and comparable to what is found for CORA-AX01 or Armor3D-AX01, except
in the central and western parts of the Irminger Sea. Although this is a region where it is
known that surface low frequency variability tends to be correlated at depth (Reverdin et
al., 2018 or the old one?), one expects a decrease of correlation between the surface and
greater depths. The better correlation with vertically integrated quantities than with
analysis at the same (5 m) level in EN4 suggests that 'noisy' or 'erroneous' data are not
properly filtered in the EN4 surface analysis. It is also found that CORA analyses are
more correlated vertically that EN4, which points in the same direction. To a large extent,
CORA and EN4 products rely on the same data, largely to Argo (and earlier PALACE)
floats as well as research cruise CTD data, whereas B-AX01 strongly relies on TSG data
(to a large extent from Nuka Arctica). These mapped products differ in how they are
produced. EN4 will tend to stick more to local data, whereas CORA analysis scheme is a
classical objective mapping of deviations from a guess field. Thus, it will damp
variability when there is not enough data within the radius of integration (Gaillard et al.,
2009). This is likely to have often been the case before the Argo float deployments in
2001-2002. Thus, CORA will underestimate the variability, but be less noisy. The lack of
data probably also explains the absence in this product of the low salinity signals in 1993-

429 1995.


For SST, the correlation of B-AX01 with all the gridded products along this zonal section
is quite large (larger than 0.80 everywhere, albeit a little smaller for Armor3D) with RMS
variability of the same magnitude to the one in B-AX01 in the different products
(although slightly smaller in CORA). The data coverage (XBTs in addition to Argo,
PALACE and CTD casts) is often quite good, with largest differences in 1993-1996 when
data coverage is weaker. Despite possible near surface stratification, the large similarity



in T between B-AX01 and EN4 suggests that the different temperature data sets are
consistent. The correlation with vertically integrated temperature is smaller than at the
surface and rather similar in the two products, again pointing to rather well data-
constrained analyses.

The comparison of TSG data with Argo profile data (App. A) gives confidence in S from
Nuka Arctica and thus in B-AX01 time series. Thus, the large difference in S between
EN4 and B-AX01 is indicative of large seasonal noise in EN4 surface salinity, maybe
resulting from the insufficient sampling of meso-scale, short-term variability, in
particular from Argo and other (earlier) profiling salinity floats. In the western and
central Irminger Sea, the objective mapping technique used in EN4 could also spread an
influence of distant data of the cold and fresh water of the east Greenland shelf and slope,
which have very different values.






**Author contribution**


GR has contributed to the data validation and data compilation along the two ship of
opportunity lines (AX01 and AX02) since the project was initiated in 1993. HV has
provided support in Iceland and contributed to the scientific discussion on the data
compilation. GA has been in charge of AX02 data correction and validation. DD has
installed the TSG on M/V Nuka Arctica in 1997 and monitored the data since then. FB
and GG at NOAA/AOML have supported the TSG and XBT operations for many years
on AX02. LC has contributed to the comparison of the gridded products to EN4, and TS
has contributed to the comparison of the gridded product with CORA. LH has been the
contact for Nuka Arctica in Nuuk (Greenland) and analyzed a large part of the water
samples used for the data calibration of AX01.

We have not identified any conflict of interest.

**Acknowledgements**
This is a contribution to the French SSS observation service, which is supported by
French agencies INSU/CNRS, IRD, CNES and IPEV. We are very grateful to the crews
of the different vessels on lines AX01 and AX02 from which the salinity data have been
collected, in particular under the EIMSKIP and Royal Arctic Line (RAL) managements.
We acknowledge the strong support to this operation by Lars Heilman in Nuuk, Magnus
Danielsen in Reykjavik, Hans Magnussen in Aalborg, Denis Pierrot and Francis Bringas.
NOAA/AOML and NOAA/CPO Ocean Observing and Monitoring Division have
contributed by maintaining the TSGs along AX02 and providing XBTs on the different
ships that have operated along the AX01 and AX02 transects. Coriolis contributed by
providing XBTs to M/V Nuka Arctica and by supporting the production of the CORA
dataset. ARMOR3D (G LOBAL_REP_PHY_001_021) products are freely available
through the Copernicus Marine Environment Monitoring service.




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



**Figure captions**
**Figure 1**. Map of the bins along B-AX01 (black), B-AX02 (red), G-AX02 (blue), and N-
AX01 (green). A typical example of ship track is shown along B-AX02.
**Figure 2**. B-AX01 (left) and B-AX02 (right) Hoevmøller diagrams of deviations from
an average seasonal cycle. Salinity (top with vertical lines indicative of the crossing),
temperature (middle), density (bottom). The sketch on top/left corner indicates
where the lines are located with relation to main currents (red NAC and extensions,
purple fresher slope and shelf currents.
**Figure 3**. G-AX01 (left) and N-AX02 (right) anomalies Hoevmøller diagrams of
deviations from an average seasonal cycle. Salinity (top), temperature (middle),
density (bottom) (see Fig. 1 for locations of sections). For salinity and density,
different contours/color codes are used for G-AX01 and N-AX01.
**Figure 4**. Seasonal cycle of interannual RMS variability (left along B-AX01; right
along B-AX02). S (top), T (middle) and density (bottom)
**Figure 5**. The principal components (PC) and spatial structure (EOF) of an empirical
orthogonal function analysis of salinity jointly for B-AX01 and B-AX02 (07/1993-
12/2017) (we applied a 15-month running mean prior to the EOF analysis). The PCs
are normalized to variance 1, and the EOF are such that 1 indicates that the EOF
explains 100% of total local variance.
**Figure B.1**. Comparison in 1993-2015 of S and T from B-AX01 with EN4 (blue) and
CORA (red) gridded data (surface, full lines; 0-500m vertically integrated, dashed
lines). Correlation coefficients are plotted, as well as the RMS standard deviations in
the different products (the dashed black line is for B-AX01 data). The upper panels
are for S, the lower panels for T.



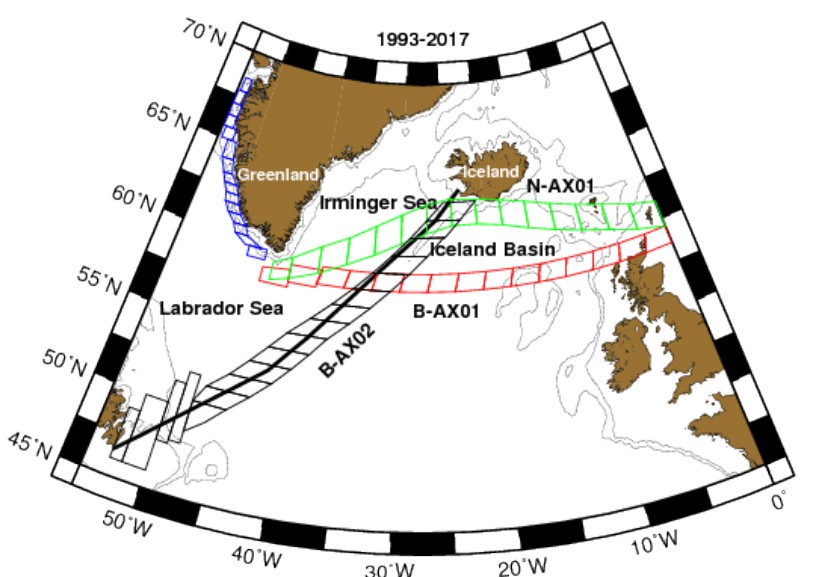

**Figure 1**. Map of the bins along B-AX01 (black), B-AX02 (red), G-AX02 (blue), and N-
AX01 (green). A typical example of ship track is shown along B-AX02.




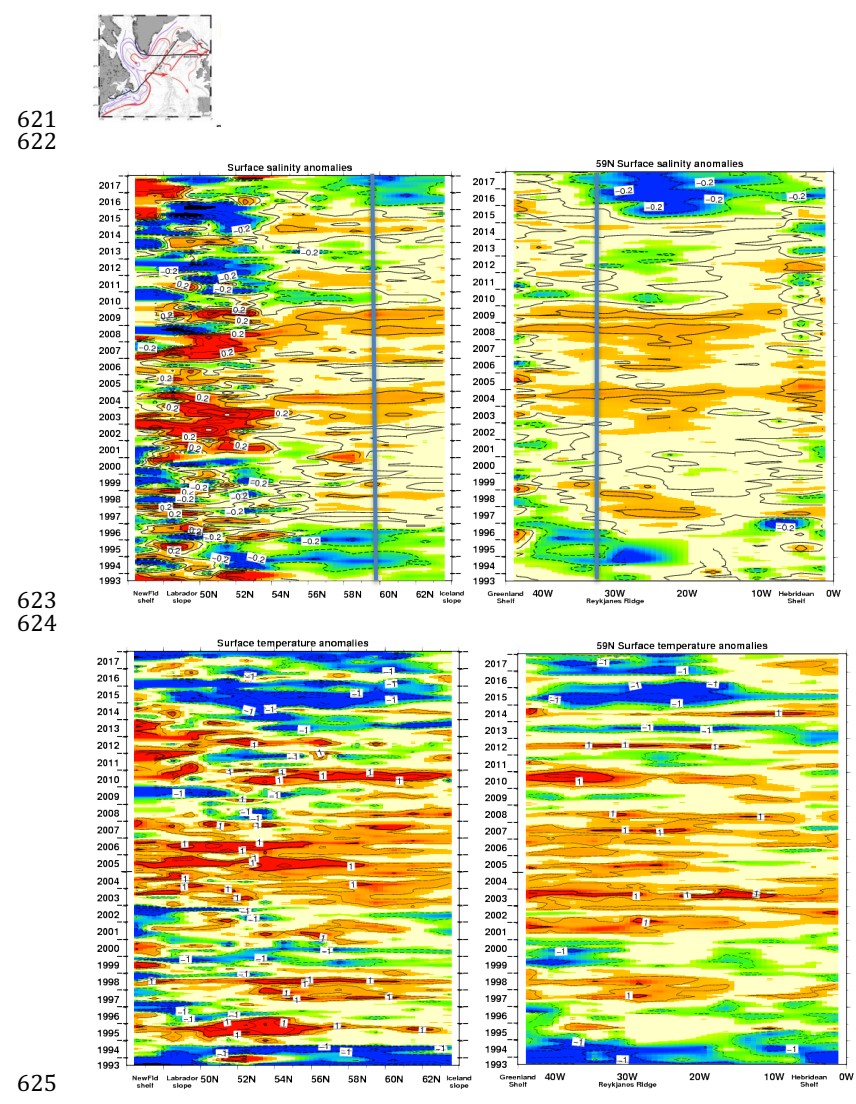







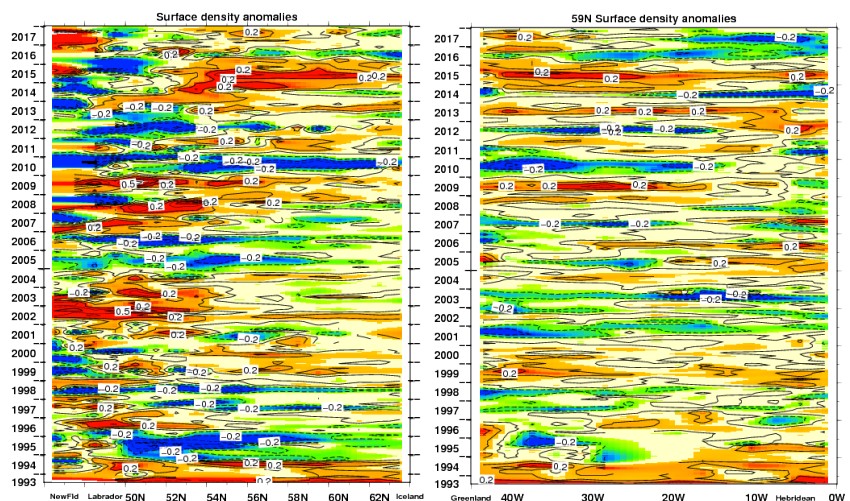

**Figure 2**. B-AX02 (left) and B-AX01 (right) Hoevmøller diagrams of deviations from
an average seasonal cycle. Salinity (top with vertical lines indicative of the crossing),
temperature (middle), density (bottom). The sketch on top/left corner indicates
where the lines are located with relation to main currents (red NAC and extensions,
purple fresher slope and shelf currents.





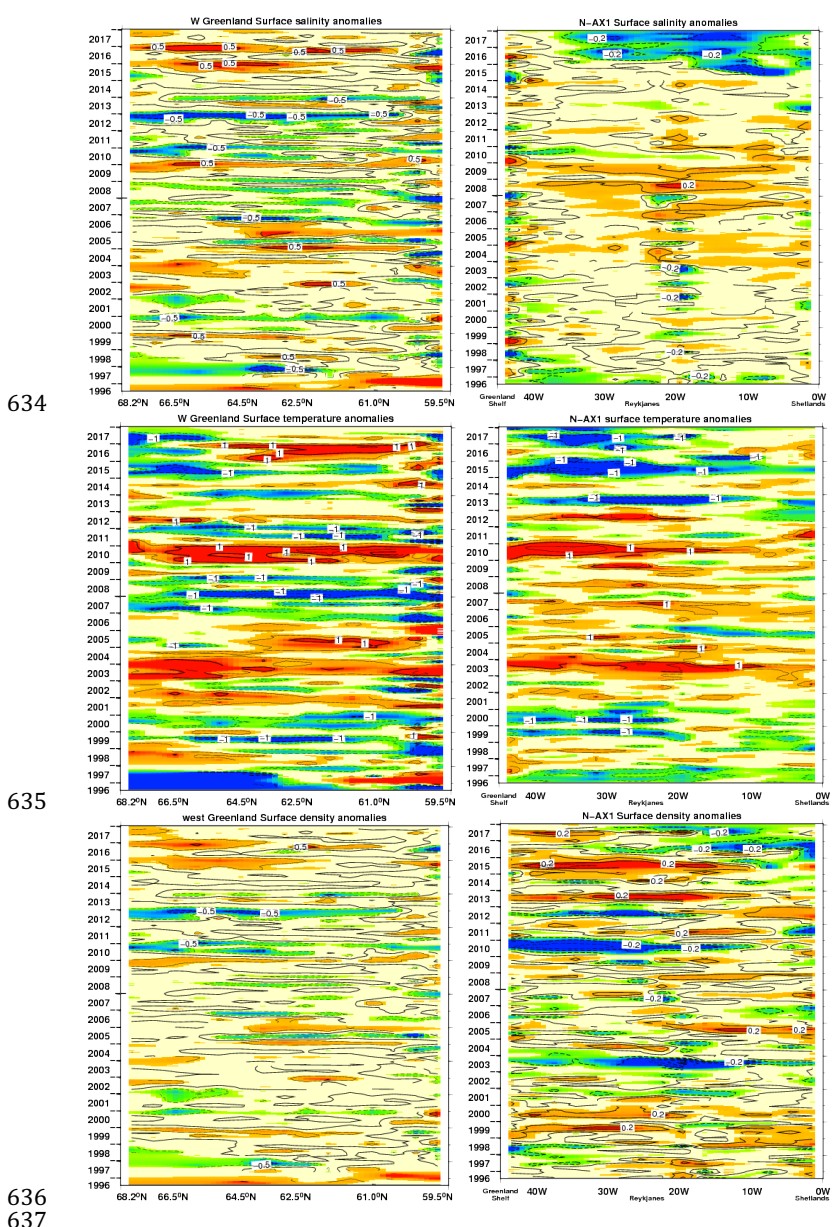

**Figure 3**. G-AX01 (left) and N-AXO1 (right) anomalies Hoevmøller diagrams of
deviations from an average seasonal cycle. Salinity (top), temperature (middle),



density (bottom) (see Fig. 1 for locations of sections). For salinity and density,
different contours/color codes are used for G-AX01 and N-AX01.






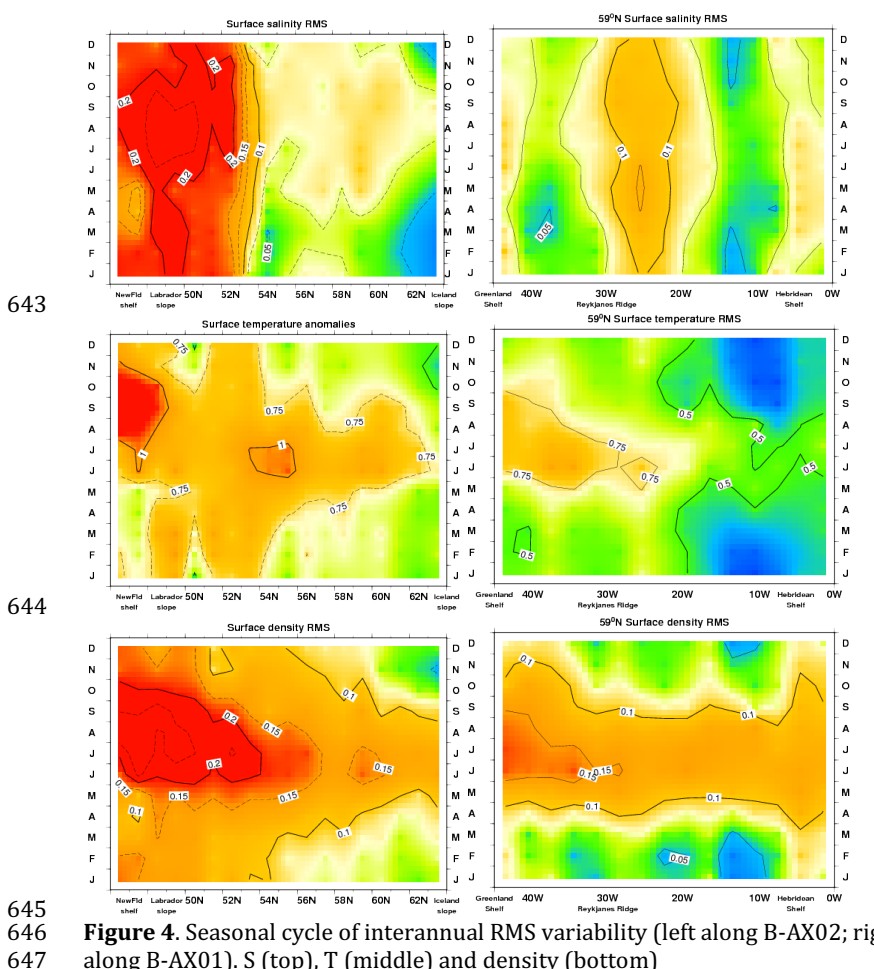

**Figure 4**. Seasonal cycle of interannual RMS variability (left along B-AX02; right
along B-AX01). S (top), T (middle) and density (bottom)







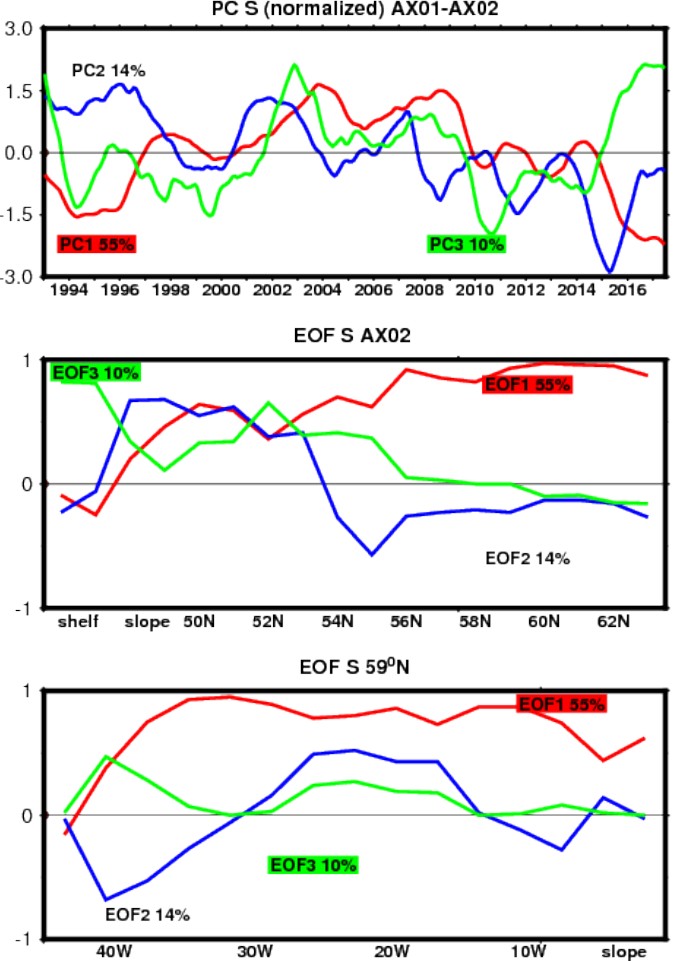

**Figure 5**. The principal components (PC) and spatial structure (EOF) of an EOF
analysis of salinity jointly for B-AX01 and B-AX02 (07/1993-12/2017) (we applied a
15-month running mean prior to the EOF analysis). The PCs are normalized to
variance 1, and the EOF are such that 1 indicates that the EOF explains 100% of total
local variance.

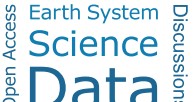




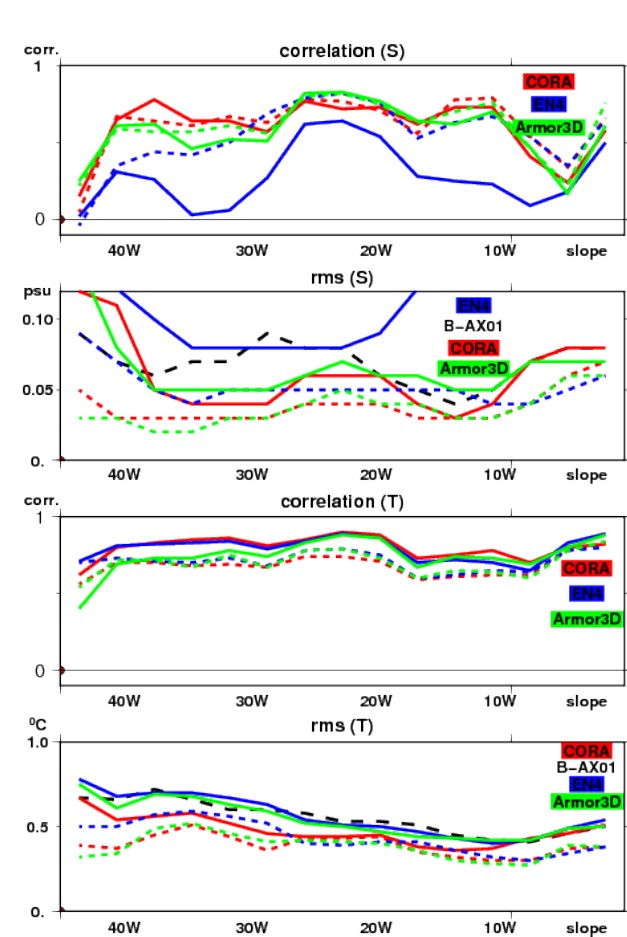

**Figure B.1**. Comparison in 1993-2015 of S and T from B-AX01 with EN4 (blue) and
CORA (red) gridded data (surface, full lines; 0-500m vertically integrated, dashed
lines). Correlation coefficients are plotted, as well as the RMS standard deviations in
the different products (the dashed black line is for B-AX01 data). The upper panels
are for S, the lower panels for T.