# Peer review of "North Atlantic subpolar gyre along predetermined ship tracks since 1993: a monthly dataset of surface temperature, salinity, and density."

_Earth System Science Data, 2018_

## Referee Comment (RC1) · Anonymous Referee #1 · 25 Apr 2018

This is an extremely useful and unique dataset, its great to see the authors publishing ship of opportunity data in a quality controlled and binned format. Overall, the presentation of the data and its context is very clear and accessible. The analysis is also sound, though the implications of the results is not thoroughly explored, which is fine given the format of the journal and the acknowledgement of the limitations of this dataset. The data itself is straightforward and easy to use. I look forward to seeing this become published and available.

Here are a few suggestions for improvement:

- The explanation of the finding the seasonal cycle (L147-151) is somewhat

opaque. The authors should clarify what "modified and adjusted" means and how the cycle connects to the Friedman et al. reference more explicitly.

• A table showing when each line was occupied and had TSG/bucket/ARGO/CTD calibration would be a useful quick reference.

• It may be useful to provide access to the full bin locations rather than just the center point.

• The abstract makes it seem as though there are only 2 transects. You may want to clarify that they have been split into several usable binned transects here.

• The statement on L45-46 "The changes in these subsurface water properties and distributions drive ocean circulation and in particular of the Atlantic Meridional Overturning Oscillation (AMOC) variability" is debatable and I would remove or qualify it with a word such as likely or making reference to the fact that wind forcing is also a large player.

• Data availability is printed in two locations (L24 and L293) not sure whether this is intentional.

• The "ad-hoc" adjustment could be clarified (L346)

• A few figures showing the comparison with Argo and other data discussed in Appendix A could be useful and strengthen the presentation.

• Figure 1: The colors are mislabeled in the caption. Should be B-AX01 (red) and B-AX02 (black) I believe. Also, you may want to label G-AX02 on the figure.

• It would be useful to see time series of the overlapping portions of the B-AX02 and B-AX01 data rather than just drawing the line through them in Figure 2.

- Figure 3: The orienting map should be made larger, made into its own figure or combined with Figure 1.

- Figure 5: The statement "the EOF are such that 1 indicates that the EOF explains 100

[Figure]

---

## Referee Comment (RC2) · Anonymous Referee #2 · 17 May 2018

This study presents newly binned datasets along four ship tracks in the subpolar North Atlantic, including monthly sea surface temperature, salinity, and density between 1993 and 2017. Based the data, this study further describes the characteristics of those key variables in the region in terms of their temporal and spatial variability. The subpolar region has been sparsely and irregularly sampled especially prior to the Argo era let alone the continental slope regions that are not available even from the Argo data which all makes this dataset based on observations from routine volunteer ships particularly interesting. However, I found some statements are rather vague and it tends to lack good transitions to connect paragraphs. Please see my main concerns along with some minor comments outlined below.

[Figure]

General comments

1. Introduction: While the first and second paraphs in Introduction make it clear why the surface observations in the subpolar region are critical, it doesn't seem to have made a good transition to the third paragraph. What is the main purpose of the third paragraph? How does it lead to the fourth paragraph? I would suggest the authors strengthen the importance and uniqueness of their data product (i.e., vs. Argo), and discuss previous studies based on the same ship data with emphasis on how the presented data could make a difference.

2. The name of the subsections is misleading. Section 3.1 actually describes variability along all the transects not just 'AX01 and AX02'. Section 3.3 discusses the interannual variability and is not about an analysis method.

3. I found a couple of places very confusing because of the lack of transparency on the method. The authors need to be specific on what data were used and how a derived quantity was defined and for what purpose. For example, what is 'seasonal cycle of interannual RMS variability'? what data were used for calculating this 'RMS variability'? Why The motivation of those analysis needs to be clarified which will help the reader to understand the present results.

Minor comments

Line 41: de Jong and de Steur 2016

Line 66-77: Please reword. See general comments above.

Line 82: Is this 'interannual standard deviation' the same as 'interannual RMS variability' used in section 3.2? Please be consistent.

Line 89: Are those cargo ships? Do they run on a regular schedule? How long does it take for sampling each transect?

Lines 149-150: It is confusing about what has been done here with the averaged seasonal cycle. Please rephrase.

Lines 150-151: What about temperature and density?

Lines 158-159: It is not clear why EOF will be performed.

Line 170: Should it be 'top left'?

Line 173: What is the error map like? What is the magnitude of the error?

Line 210: Since N-AX01 is a variation of B-AX01 and has been less frequently sampled, I would suggest sticking with the B-AX01 for discussions. Then Figures 1 and 2 can be merged. Figure 2 right panel on N-AX01 can be moved to Supplementary.

Line 236ff: It is not clear what data were used for calculating those RMS values – still monthly anomalies that have the seasonal cycles removed? Also, what about the comparison between AX01 and GX01?

Line 237: Please define 'RMS'.

Line 275: Fig. 5.

Line 302: 'the seasonal variability'? The seasonal cycles have been removed from the data for analysis. . .

Line 415: 'or the old one?'

Line 419: Typo 'than'.

Line 592: B-AX02 (left) and B-AX01 (right).

Figure 1: B-AX01 (red) and B-AX02 (black). What is G-AX02? It doesn't appear in the text.

Figures 2: The sketch on the top left corner is too hard to read.

Figures 2 and 3: Please put the name of the transect in the figure titles and use 'B-AX01' instead of '59 N' for consistency. Please add the colorbar.

Figure 4: Please add the colorbar.

Figure 5: Also, please replace '59 N' with AX01 for consistency.

Data files:

1. The global attribute 'title' in each file should contain the name of the ship track (e.g., AX01). Then the user can find any relevant studies used the data collected along that transect.

2. Error information seems to be missing for each variable included in the data file.
* * *

---

## Author Comment (AC1) · 26 Jun 2018

Response to the two reviews
We thank the reviewers for their positive comments and their suggestions.
We provide below a detailed response to the queries, and how we modified the manuscript.

**Anonymous Referee #1**

This is an extremely useful and unique dataset, its great to see the authors publishing ship of opportunity data in a quality controlled and binned format. Overall, the presentation of the data and its context is very clear and accessible. The analysis is also sound, though the implications of the results is not thoroughly explored, which is fine given the format of the journal and the acknowledgement of the limitations of this dataset. The data itself is straightforward and easy to use. I look forward to seeing this become published and available.
Here are a few suggestions for improvement:
    • The explanation of the finding the seasonal cycle (L147-151) is somewhat opaque. The authors should clarify what "modified and adjusted" means and how the cycle connects to the Friedman et al. reference more explicitly.
Authors: same comment as referee 2. Thank-you, we have clarified how we proceeded.

• A table showing when each line was occupied and had TSG/bucket/ARGO/CTD calibration would be a useful quick reference.
Authors: interesting suggestion, however, the table would be complicated, and we prefer to explain it in the text. Furthermore, a table is now added in Appendix A that summarizes the comparisons done (TSG with Argo and XBT data).

• It may be useful to provide access to the full bin locations rather than just the center point.
Authors: This is now provided on the WEB site.

• The abstract makes it seem as though there are only 2 transects. You may want to clarify that they have been split into several usable binned transects here.
Authors: thank-you. We have rewritten this part of the abstract.

• The statement on L45-46 "The changes in these subsurface water properties and distributions drive ocean circulation and in particular of the Atlantic Meridional Overturning Oscillation (AMOC) variability" is debatable and I would remove or qualify it with a word such as likely or making reference to the fact that wind forcing is also a large player.
Authors: we added 'plays a likely role in'. We also believe that wind forcing is a key player.

• Data availability is printed in two locations (L24 and L293) not sure whether this is intentional.
GR: Verifier si on peut retirer l. 24
• The "ad-hoc" adjustment could be clarified (L346)
Authors: we removed 'ad hoc' and rephrased the statement. What we meant is that a small adjustment was done only when the nearby data suggested a non-zero difference which was not common.

• A few figures showing the comparison with Argo and other data discussed in Appendix A could be useful and strengthen the presentation.

Authors: this is an interesting suggestion. On the other hand, the comparisons are at first order summarized by the average and rms standard deviations, and the size of the set of comparisons is usually not sufficient to investigate higher moments of the distributions.
We thus added a table summarizing these comparisons.

• Figure 1: The colors are mislabeled in the caption. Should be B-AX01 (red) and B-AX02 (black) I believe. Also, you may want to label G-AX02 on the figure.
Authors: thank-you. Corrected and we labeled G-AX01 on the figure.

• It would be useful to see time series of the overlapping portions of the B-AX02 and B-AX01 data rather than just drawing the line through them in Figure 2.
Authors: by construction, the time series for the overlapping portions of B-AX02 and B-AX01 are quite close, as they share a good part of the data, a similar method to construct them and a very close seasonal cycle (Figure 1 indicates a large overlap of the bins). Small differences occur because of 'gap-filling' in the time series of B-AX01and a larger amount of meridional smoothing. For this common box, the difference signal has average amplitude of (0.27, 0.030) in T and S, compared to an average signal amplitude of (0.54, 0.068). The difference is within the range of the error estimates, but as they share a large part of the data, it is not straightforward to interpret, and a figure would not help.

• Figure 3: The orienting map should be made larger, made into its own figure or combined with Figure 1.
Authors: We suspect that this refers to Figure 2. We agree that this panel is too small, and removed it, as Figure 1 provides the information on the binning. Titles on panels of figure 2 now also refer directly to the names of the lines on Figure 1.

• Figure 5: The statement "the EOF are such that 1 indicates that the EOF explains 100
Authors: Unfortunately, the end of the referee's comment is missing. Our statement might not have been very clear, and we removed 'variance'

**Anonymous Referee #2**

This study presents newly binned datasets along four ship tracks in the subpolar North Atlantic, including monthly sea surface temperature, salinity, and density between 1993 and 2017. Based the data, this study further describes the characteristics of those key variables in the region in terms of their temporal and spatial variability. The subpolar region has been sparsely and irregularly sampled especially prior to the Argo era let alone the continental slope regions that are not available even from the Argo data which all makes this dataset based on observations from routine volunteer ships particularly interesting. However, I found some statements are rather vague and it tends to lack good transitions to connect paragraphs. Please see my main concerns along with some minor comments outlined below.

General comments
1. Introduction: While the first and second paragraphs in Introduction make it clear why the surface observations in the subpolar region are critical, it doesn't seem to have made a good transition to the third paragraph. What is the main purpose of the third paragraph? How does it lead to the fourth paragraph? I would suggest the authors strengthen the importance and uniqueness of their data product (i.e., vs. Argo), and discuss previous studies based on the same ship data with emphasis on how the presented data could make a difference.

Authors: thank-you to point that out. We have followed the advice, restructured paragraphs 2 to 4 and have developed the presentation of the data product, as well as its uniqueness in comparison with Argo (the time and space resolution, getting onto the shelves…) and earlier published studies we did with these data (only on B-AX02).

2. The name of the subsections is misleading. Section 3.1 actually describes variability along all the transects not just 'AX01 and AX02'. Section 3.3 discusses the interannual variability and is not about an analysis method.
Authors: we have renamed the sections

3. I found a couple of places very confusing because of the lack of transparency on the method. The authors need to be specific on what data were used and how a derived quantity was defined and for what purpose. For example, what is 'seasonal cycle of interannual RMS variability'? what data were used for calculating this 'RMS variability'? Why The motivation of those analysis needs to be clarified which will help the reader to understand the present results.
Authors: The standard deviation of interannual variability (RMSa) is estimated using the monthly time series of deviations from the average seasonal cycle. It has one value for each calendar month which is estimated from the 25 (or less) 'anomalies' (in different years) of the time series for that calendar month. This is why RMSa portrays a seasonal cycle. Surface variability is expected to be seasonally modulated (surface forcing, mixed layer depth…). Thus knowing to which extent this parameter exhibits a seasonal variability is very important  to comprehend the seasonal behavior of the time series and needs to be presented.

Minor comments
Line 41: de Jong and de Steur 2016
Authors: corrected

Line 66-77: Please reword. See general comments above.
Authors: thank-you. Sections 2 to 4 of the introduction have been reorganized, following the suggestion of the reviewer.

Line 82: Is this 'interannual standard deviation' the same as 'interannual RMS variability' used in section 3.2? Please be consistent.
Authors: Thank-you. We now use 'standard deviation of interannual variability' throughout the paper (we refer to it as RMSa)

Line 89: Are those cargo ships? Do they run on a regular schedule? How long does it take for sampling each transect?
Authors: These are cargo ships, usually running on a regular schedule, with crossings from shelf to shelf on the order of 4 days both for AX01 and AX02. Unfortunately, for various reasons (storms, changes of ships, contracts…), it is not as regular as hoped for and the repeatability of a section can vary from two weeks to a few months. This was complemented (mostly along AX02) by TSG data from a few research cruises.

Lines 149-150: It is confusing about what has been done here with the averaged seasonal cycle. Please rephrase.
Lines 150-151: What about temperature and density?
Authors: A similar comment was made by reviewer 1. We adopted the same approach for temperature as for salinity, whereas the density time series are estimated from the temperature and salinity time series, and not from individual density data first binned, as described for T and S, thus a slightly different approach. This has now been clarified, and the section has been rewritten.

Lines 158-159: It is not clear why EOF will be performed.
Authors: The EOF analysis is performed to highlight the value of the data presented in this study. It demonstrates that the variability extracted from the joint bins along B-AX01 and B-AX02 captures key physical processes regulating the climate of the subpolar gyre. However, as the Reviewer may have noticed, we briefly point this out in the paper without going into any details.

Line 170: Should it be 'top left'?
Authors: yes. B-AX01 and B-AX02 panels had been inverted on Figure 2.

Line 173: What is the error map like? What is the magnitude of the error?
Authors: Error estimates vary greatly from month to month due to changes in the sampling and smaller scale variability. They are typically larger where variability is largest, and normalized to the signal amplitude, they are largest in the slope regions (because of insufficient sampling in the presence of large small scale variability). For example for S in the westernmost box and the two easternmost boxes of B-AX01 as well as for the two slope boxes (3 and 4 counted from the south) on B-AX02, the average ratio of error/signal is larger than 0.4, whereas it is less than 0.3 in most boxes. Interestingly, it becomes also larger than 0.4 for three eastern boxes of B-AX01 (11-13), where the signal has small amplitude. Although error estimates are interesting in themselves, we don't think that this could be presented in an easily comprehensive way. Instead, we will now provide them with the binned data.

Line 210: Since N-AX01 is a variation of B-AX01 and has been less frequently sampled, I would suggest sticking with the B-AX01 for discussions. Then Figures 1 and 2 can be merged. Figure 2 right panel on N-AX01 can be moved to Supplementary.
Authors: Indeed, there were times in particular before 1997, when N-AX01 was less frequently sampled (which also indicated by larger error estimates). This is why we chose (as for G-AX01) to start its time series in 1996. We agree that this is somewhat redundant with B-AX01, and only discussed a few differences between the two sections. However, removing N-AX01 does not save space as G-AX01 has also a reduced time span (thus also, not the same period over which the climatological seasonal cycle is estimated) and cannot easily be merged with Figure 2 (B-AX01 and B-AX02). Thus, we prefer to leave it in the core of the paper.

Line 236ff: It is not clear what data were used for calculating those RMS values – still monthly anomalies that have the seasonal cycles removed? Also, what about the comparison between AX01 and GX01?
Authors: We agree that the wording was confusing. We are indeed considering deviations from the seasonal cycle, and are investigating the rms stardard deviations of these deviations for the different calendar months (thus building a seasonal cycle of it). The title of section 3.2 has also been changed.

Line 237: Please define 'RMS'.
Authors: done

Line 275: Fig. 5.
Authors: thank-you.

Line 302: 'the seasonal variability'? The seasonal cycles have been removed from the data for analysis: : :
Authors: This is now clarified. The wording should be 'variability on seasonal time scales', instead of 'seasonal variability'.

Line 415: 'or the old one?'
Authors: 'or the old one' has been removed.

Line 419: Typo 'than'.
Authors: corrected

Line 592: B-AX02 (left) and B-AX01 (right).
Authors: thank-you. Corrected.

Figure 1: B-AX01 (red) and B-AX02 (black). What is G-AX02? It doesn't appear in the text.
Authors: Thank-you. G-AX02 should have been G-AX01 and refers to the sampling along west Greenland. The label has been added on the figure.

Figures 2: The sketch on the top left corner is too hard to read.
Authors: We agree and it is redundant with Fig. 1. Thus, it was removed.

Figures 2 and 3: Please put the name of the transect in the figure titles and use 'BAX01' instead of '59 N' for consistency. Please add the colorbar.
Authors: We changed the titles and added a colorbar

Figure 4: Please add the colorbar.
Authors: we added a colorbar

Figure 5: Also, please replace '59 N' with AX01 for consistency.
Authors: we replaced it by B-AX01 for consistency (and changed AX02 in B-AX02)

Data files:
1. The global attribute 'title' in each file should contain the name of the ship track (e.g. AX01). Then the user can find any relevant studies used the data collected along that transect.
Authors: we followed this advice

2. Error information seems to be missing for each variable included in the data file.
Authors: We agree that this was missing and might be interesting information for the user. It was estimated directly for T and S for B-AX02, G-AX01 and N-AX01. It is a little bit more complicated for B-AX01, where some periods with missing data were filled with nearby bins (in latitude) data (near 58°N and 60°N respectively). In these cases, the error is estimated to be the largest one of the different bins with data. Then, error is estimated for density, derived from the errors in T and S time series when estimating density (assuming that these errors are independent).